# Silica In Silico: A Molecular Dynamics Characterization of the Early Stages of Protein Embedding for Atom Probe Tomography

**Giovanni Novi Inverardi** [1,2], **Francesco Carnovale** [1,2], **Lorenzo Petrolli** [1,2], **Simone Taioli** [2,3,4] **and Gianluca Lattanzi** [1,2,*]

1   Department of Physics, University of Trento, Via Sommarive 14, 38123 Povo-Trento, Italy; lorenzo.petrolli@unitn.it (L.P.)

2   INFN-TIFPA Trento Institute for Fundamental Physics and Applications, Via Sommarive 14, 38123 Povo-Trento, Italy

3   European Centre for Theoretical Studies in Nuclear Physics and Related Areas (ECT*), Bruno Kessler Foundation, 38123 Trento, Italy

4   Faculty of Applied Physics and Mathematics, Gdańsk University of Technology, 80-309 Gdańsk, Poland

\*   Correspondence: gianluca.lattanzi@unitn.it; Tel.: +39-0461-283104

**Abstract:** A novel procedure for the application of atom probe tomography (APT) to the structural analysis of biological systems, has been recently proposed, whereby the specimen is embedded by a silica matrix and ablated by a pulsed laser source. Such a technique, requires that the silica primer be properly inert and bio-compatible, keeping the native structural features of the system at hand, while condensing into an amorphous, glass-like coating. In this work, we propose a molecular dynamics protocol, aimed at depicting and characterizing the earliest stages of the embedding process of small biomolecules in a solution of water and orthosilicic acid, here, taken as a precursor of the silica matrix. Overall, we observe a negligible influence of orthosilicic acid on the behavior of stable folded systems (such as ubiquitin). Conversely, intrinsically disordered and unstable peptides are affected by the coating, the latter seemingly inhibiting the fluctuations of flexible moieties. While further scrutiny is in order, our assessment offers a first mechanistic insight of the effects of orthosilicic acid, thereby validating its use in the proposed innovative application of APT to the structural resolution of protein molecules.

**Keywords:** molecular dynamics simulations; protein embedding; atom probe tomography; orthosilicic acid

## 1. Introduction

The characterization of the correlations between structural features and functional roles of biomolecular systems, is a crucial theme in molecular biology. Traditionally, the reconstruction of the three-dimensional structure of biomolecules at a sub-nanoscale resolution, has been mostly achieved by means of non-invasive techniques such as X-ray crystallography [1], nuclear magnetic resonance (NMR) [2], and, more recently, by cryogenic electron microscopy (cryo-EM) [3]. Yet, an alternative approach, mostly employed in the analysis of metals and alloys, is offered by atom probe tomography (APT), whose range of applications has broadened in recent decades [4]. The fundamental principle of APT, is based on a spatially resolved mass-spectrometry technique: a biological specimen is embedded by a needle-shaped matrix (radius < 100 nm) and its individual atoms are forcefully ejected via the application of a strong electric field (over $1\,\mathrm{V} \times \mathrm{Å}^{-1}$), upon illumination by short laser pulses.

A novel APT-based pipeline has been recently proposed [5], improving on an earlier embedding strategy [6]. In brief, a solution of the biological system and of a silica precursor is dried steadily, thereby achieving, first, a gel-like and, subsequently, a glassy phase, upon condensation of an amorphous silica matrix. An in situ lift-out procedure is thus performed,

by means of a focused ion beam-scanning electron microscope (FIB–SEM) [7], sharpening the matrix into the shape of a needle, which is analyzed by APT.

In [5], several embedding protocols are described, the most relevant for the purpose of this work, concerns the use of orthosilicic acid ($Si(OH)_4$) as a precursor of the silica matrix. $Si(OH)_4$ offers significant advantages, namely, (i) it is readily available in the form of commercial sodium ortho-silicate ($Na_4SiO_4$) solutions, and (ii) it spontaneously (albeit slowly) self-condenses at room temperature, into nano-sized aggregates of 1 mM, eventually yielding a glassy matrix upon evaporation of the water solvent [8]. This process might be enhanced by in vitro techniques, such as prolonged curing at 37° for over 48 hours [5]. This experimental setup has been employed to resolve the tertiary structure of a rabbit immunoglobulin G (IgG), at a resolution below 15 Å.

In principle, this novel APT-based protocol might be applied as an alternative to conventional experimental techniques, to capture, for instance, the conformational flexibility of intrinsically disordered systems. However, a few critical aspects deserve a careful evaluation. Firstly, IgG antibodies are massive structures, characterized by a high stability towards the mechanical stresses arising from encapsulation within the silica matrix. In this concern, the applicability of the embedding procedure should thus be assessed over smaller, more flexible systems. On the other hand, the coating by silica, suppresses local motions due to thermal fluctuations, the latter accounting for a loss in spatial and chemical resolution, while keeping the functional, native structure of the system. Indeed, it was experimentally observed that encapsulated enzymes retain their catalytic activity [9], thereby inferring that silica precursors behave as inert materials.

In this context, in silico, numerical approaches, such as molecular dynamics (MD) [10], might be instrumental in depicting the different stages of the embedding process, characterizing the interactions between the biological systems and the silica matrix. MD requires accurate, robust, and versatile models of the inter-atomic forces or force fields: While several atomistic models for biomolecules have been developed in recent decades, no general force field applicable to organosilicon compounds and silicates was available until recently. Indeed, there have been several attempts at developing general models describing solid-state, silica-based materials [11,12]; however, a unique, robustly validated force field for organic orthosilicates in solution, namely PolCA (polarization–consistent approach), was first achieved by Jorge et al. in 2021 [13]. Inspired by theoretical developments in the approximate treatment of polarization effects, PolCA describes a broad variety of orthosilicate derivatives, involving bonded and non-bonded additive interactions, refined upon thermodynamic and electronic experimental data.

Here, we implement a preliminary workflow aimed at characterizing the interactions between $Si(OH)_4$ (taken as the precursor of the embedding silica matrix) and a set of small proteins and peptides, displaying diverse structural features, at the molecular scale. Namely, the influence of orthosilicic acid on the behavior of ubiquitin, of the small ubiquitin-like modifier 1 (SUMO-1), and of the transmembrane helix of the interleukin-22 receptor alpha-1 (IL22R$\alpha$1), is assessed by means of classic atomistic MD simulations, employing the ff99sb-ildn Amber force field for proteins, in combination with the PolCAFF of silicate derivatives. Ubiquitin is associated with a well-characterized and stable conformation in solution, and is here employed as a putative benchmark scenario, to broadly assess the role of $Si(OH)_4$. SUMO-1 is equipped with a ubiquitin-like core and intrinsically disordered carboxy and amino-termini, whose behavior is characterized in terms of their inherent flexibility. Lastly, IL22R$\alpha$1 is here assessed as representative of an unstable, purely hydrophobic peptide in solution. Overall, we observe the early onset of the embedding process, arguably driven mostly by entropic effects rather than site-specific interactions, and draw preliminary inferences on the capability of orthosilicic acid to stabilize the dynamics of small biological systems.

## 2. Materials and Methods

### 2.1. System Structures and Setup

The equilibrium coordinates of Si(OH)$_4$ were retrieved by DFT calculations on the Quantum Espresso package [14,15], employing a plane wave expansion of the electronic wave functions and charge density. The exchange and correlation functional was treated with the generalized gradient approximationm as formulated by Perdew, Burke, and Ernzerhof [16], associated with the optimized norm-conserving Vanderbilt pseudopotentials [17]. Cutoffs of 100 and 400 Ry were chosen for the kinetic energy and the charge density, respectively, to achieve a convergence threshold on the total energy of the system of about 1 meV/atom. A vacuum buffer larger than 15 Å was employed in all calculations, to avoid spurious periodic interactions.

The equilibrium structures of ubiquitin and of SUMO-1 were retrieved from the 1UBQ [18] and 1A5R [19] entries in the RCSB Protein Data Bank [20], respectively. The 21-residue sequence of the transmembrane helix of IL22Rα1 [21], added with short overhangs at both termini, depicting its native chemical environment (full sequence: DRTWTYSF-SGAFLFSMGFLVAVLCYLSYRYV), was taken from the UniProt database [22], and the structure of the helix was built with UCSF Chimera [23] (the full IL22Rα1 sequence is found in [21]). Moreover, the amino and carboxy extremities of the helix were capped by an acetyl and a methylamine group, respectively, employing the PyMOL software [24], to exclude electrostatic interactions between the charged termini of the helix.

A water–silica system was assembled, by stochastically placing 200 replicas of the equilibrated Si(OH)$_4$ molecule, homogeneously distributed within an 80 Å cubic simulation box. The system was thus solvated by TIP3P water molecules [25] and neutralized by an excess 150 mM (physiological) concentration of sodium chloride.

Three control systems involving ubiquitin, SUMO-1, and the transmembrane helix of IL22Rα1, were separately built from their equilibrium structures. In addition, three water–silica, *test* setups were obtained, by placing each biomolecule within a cubic box defined upon the last frame of the Si(OH)$_4$ trajectory, employing the VMD 1.9.3 interface [26]. Likewise, all systems were solvated at a physiological (150 mM) concentration of sodium chloride.

### 2.2. MD Protocol and Force Field

The parameters for the Si(OH)$_4$ molecule were derived from PolCAFF [13], while the ff99sb-ildn Amber force field [27], associated with the Joung–Cheatham treatment of sodium chloride [28], was employed to describe all systems assessed in this work (ubiquitin, SUMO-1, IL22Rα1).

Classic MD simulations were performed, employing Gromacs 2018 [29]. Prior to raw data collection, energy minimization of the biomolecular systems, as well as of the Si(OH)$_4$ molecules, was carried out, by a steepest descent algorithm, upon convergence of the interatomic forces below 100 kJ/(mol·nm). All systems were thus equilibrated in the NVT ensemble, at T = 298 K for 1 ns, employing a Berendsen thermostat, and subsequently in the NPT ensemble, at P = 1 bar for 1 ns, via a Parrinello–Rahman barostat [30]. In both thermalization steps, mild restraints were applied to all heavy atoms of the solute molecules, associated with force constants of 1000 kJ/(mol·nm). With the exception of SUMO-1, which was simulated for 1 μs, unrestrained, 500 ns MD trajectories were carried out for each system in the NPT ensemble, associated with an integration step of 1 fs.

### 2.3. Analysis Protocol

The trajectory and timeline tools of VMD [26] were employed for the analysis of the RMSD, and of the RMSF and secondary structure content, respectively. Hydrogen bonds were evaluated applying a cutoff distance of 3.5 Å between the donor and the acceptor moieties, and a tolerance of 25 degrees. The Gromacs implementation of the algorithm developed by Eisenhaber et al. [31], was employed to estimate the solvent-accessible surface area (SASA) of ubiquitin, associated with a threshold radius of 1.4 Å.

The MDAnalysis [32] and SciPy Python libraries [33] were employed to extract significant conformations of the biomolecular systems, and to carry out a hierarchical clustering assessment on the dynamics of the transmembrane helix of IL22Rα1, respectively. As for the latter, an unweighted average linkage criterion was adopted, which recalculates the metrics upon the average over all the elements belonging to the newly achieved clusters.

An in-house, adapted version of the PyInteraph toolkit [34,35], which is built on MDAnalysis libraries, was employed to track the occupancy of the intermolecular contacts between residues of the ubiquitin-like core of SUMO-1. Electrostatic contacts—or salt bridges—in PyInteraph are established between (any atom of) an acidic and a basic moiety lying within 4.5 Å; hydrophobic contacts are effective between the center of mass of hydrophobic side chains within 5 Å; hydrogen bond criteria are satisfied as described earlier in the text.

The protein angular dispersion ($PAD_\omega$) [36], whereby we characterize the plasticity of each residue belonging to the unstructured domains of SUMO-1, is defined as:

$$PAD_\omega = \frac{180}{\pi} \cos^{-1} \left( \frac{1 - CS_\omega}{1 + CS_\omega} \right) \tag{1}$$

with $CS_\omega$ the circular spread of the angle $\omega$, the latter being sum of the Ramachandran angles $\omega = \phi + \psi$:

$$CS_\omega = \frac{1 - R_{2\omega}}{2R_{1\omega}^2} \tag{2}$$

here, $R_{1\omega}$ and $R_{2\omega}$ are the lengths of the first- and second-order sample moments of the distribution of $\omega$, respectively, defined as:

$$R_{k\omega} = \frac{1}{N} \left| \sum_{j=1}^{N} e^{ik\omega_j} \right| ; \quad k = 1,2. \tag{3}$$

with the summation running over the $j = 1$ to $N$ unit vectors $e^{ik\omega_j}$, sampled from an MD trajectory. $PAD_\omega$ describes the flexibility of a residue within a range of 0° (no dispersion in the values of $\phi$ and $\psi$) to 180° (stochastic distribution of $\phi$ and $\psi$), properly quantifying the spread of the backbone angles across a protein sequence.

## 3. Results

### 3.1. MD Assessment of the Water–Silica System

We firstly characterized the equilibrium dynamics of a water–silica medium, thereby establishing a benchmark to our subsequent (ubiquitin, SUMO-1, IL22Rα1) assessments. The system is over-saturated with silica molecules, i.e., the concentration of $Si(OH)_4$ is approximately 600 mM, about the order of magnitude of an APT experimental setup.

Despite employing a somewhat high concentration of $Si(OH)_4$, the molecules of orthosilicic acid show scarce (if at all) tendency to self-aggregate within the time scales sampled in this work—that is, about 500 ns. Indeed, the radial distribution function ($g(r)$) of the silicon atoms depicts a broadly converging trend (Figure 1A), associated with a homogeneous distribution of (mostly) non-interacting $Si(OH)_4$ molecules. This is corroborated by the individual hydrogen bonds between $Si(OH)_4$ molecules never exceeding a 1% occupancy, over the total amount of trajectory frames.

In addition, the radial distribution function of the oxygen atoms of water about the silicon atoms of $Si(OH)_4$ (Figure 1B), describes a layout of ordered solvation shells peaking at 3.5 Å: This is associated with a first solvation layer of 10 water molecules about each molecule of orthosilicic acid (as shown by the shaded area and the solid blue line in Figure 1B, depicting the cumulative value of the $g(r)$), which is in line with $Si(OH)_4$ being (moderately) soluble in water.

A
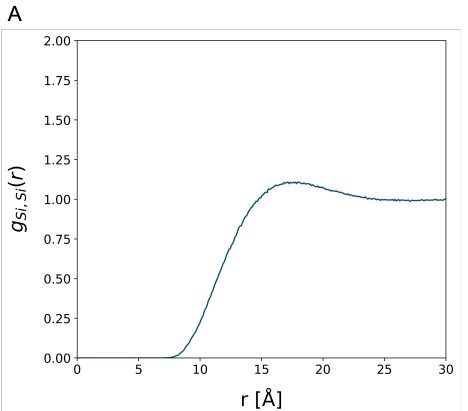

B
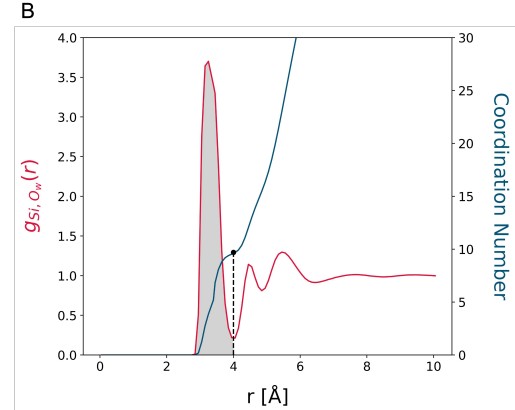

**Figure 1.** (**A**) Radial distribution function ($g(r)$) between silicon atoms of the water–silica system, depicting a broad, homogeneous distribution of Si(OH)$_4$ molecules. (**B**) Radial distribution function of the oxygen atoms of water about the silicon atoms of Si(OH)$_4$ (red line), displaying a peak about 3.5 Å. The blue line describes the cumulative distribution of the $g(r)$, with the shaded area under the peak quantifying the coordination number of Si(OH)$_4$—that is, about 10 water molecules per molecule of orthosilicic acid.

To verify whether the sampled dynamics of the water–silica system are effectively convergent, we subdivided the trajectory into 25 ns blocks and calculated the Jensen–Shannon divergence, defined as:

$$\text{JSD}(P||Q) = \tfrac{1}{2}\,\text{KL}(P||M) + \tfrac{1}{2}\,\text{KL}(Q||M) \tag{4}$$

between the (P, Q) normalized radial distribution functions associated with different blocks of the trajectory. Here, M = $\tfrac{1}{2}(P + Q)$, and KL is the Kullback–Leibler divergence between discrete distributions:

$$\text{KL}(Q||M) = \sum_r Q(r) \log\left(\frac{Q(r)}{M(r)}\right) \tag{5}$$

with $r$ the inter-atomic distance, sampled with a 0.133 Å stride. Indeed, Figure 1 in the Supplementary Data, showing the two-dimensional matrix of the JSD values, confirms the picture of rapidly converging dynamics.

### 3.2. MD Assessment of Ubiquitin

Ubiquitin is associated with a well-characterized, stable structure in solution, involving two main structural domains, namely a $\beta$-sheet and an $\alpha$-helix [18]. In line with the overall purpose of our assessment, we thus verified whether the addition of Si(OH)$_4$ somehow affects the conformational and dynamical stability of ubiquitin in solution, within the timescales sampled in this work.

Notably, the addition of Si(OH)$_4$ scarcely (if at all) interferes with the overall conformational stability of ubiquitin. Indeed, Figure 2A shows small differences in the root mean square fluctuations of the residues about the average structures of ubiquitin extracted from either trajectory (Figure 2B), and displaying a high similarity in the two scenarios. We note that, both RMSF trends match the experimental beta factors associated with the $\alpha$-carbons in the ubiquitin sequence [18] (shown by a dashed line in Figure 2A). These results tally with those reported in [5], implying that the orthosilicic acid interacts poorly with the solutes, both prior and throughout the polymerization of the silica matrix, the former being a critical stage of the APT protocol.

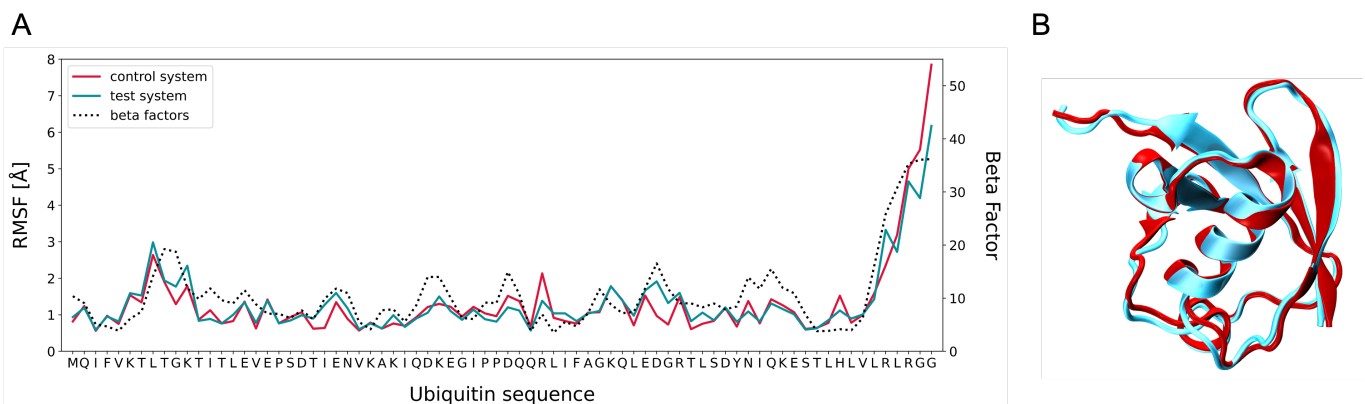

**Figure 2.** (**A**) RMSF and (**B**) average structure of the *control* (red) and *test* (blue) ubiquitin systems, displaying a high similarity between the dynamics of the two investigated scenarios. Both RMSF trends match the experimental beta factors of the $\alpha$-carbons in the ubiquitin sequence (dashed line in (**A**)).

Nevertheless, a coating shell of $Si(OH)_4$ molecules about ubiquitin, is observed in our simulation—40 to 50 molecules on average, within 5 Å of the surface of ubiquitin—accounting for an effective 20–30% loss of solvent-accessible surface area in the *test* scenario (shown in Figure S2 in the Supplementary Data). Here, no significant contribution to the observed decrease in the SASA is attributable to (local) conformational rearrangements of ubiquitin by $Si(OH)_4$.

Th coating shell does not seemingly emerge from specific interactions between ubiquitin and orthosilicic acid: we tracked the occupancy of all transient hydrogen bonds between the silanol groups of $Si(OH)_4$ and ubiquitin, observing that these interactions are mostly labile in nature but for a few exceptions exceeding 10% of the simulation frames. In this concern, we identified a small cavity on the surface of ubiquitin, seamlessly hosting two $Si(OH)_4$ molecules over a major fraction of the trajectory, which mostly involves hydrophilic and/or charged residues—namely LYS27, ASP39, GLN41, LEU43, GLN49, LEU50, GLU51, ARG72, and ARG74 (highlighted in blue in Figure 3). Within this niche, the hydrogen bonding donor and acceptor moieties are quickly exchanged between the molecules of orthosilicic acid and the neighboring amino acids; yet, most interactions are associated with an occupancy between 15 and 32% of the trajectory frames, with $Si(OH)_4$ often establishing multiple hydrogen bonds simultaneously.

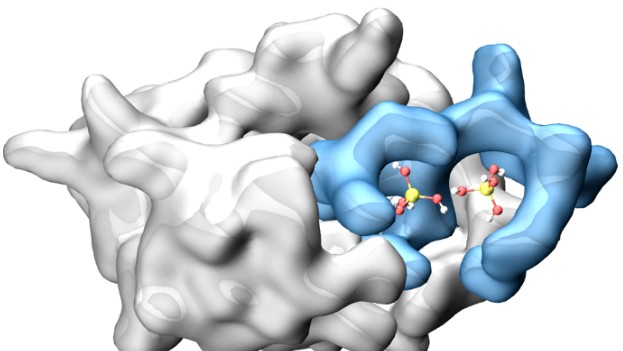

**Figure 3.** The structure of ubiquitin, depicted as an electron density isosurface: The cavity on the surface of ubiquitin, seamlessly hosting two molecules of orthosilicic acid in the *test* scenario, is highlighted in blue (details in the text).

The emergence of an embedding shell around the surface of ubiquitin, favors the interaction between molecules of orthosilicic acid, as implied by the increase in transient hydrogen bonds between silanol groups (shown in Figure S3 of Supplementary Data).

Arguably, this effect, as well as the existence of a surface cavity intercalating $Si(OH)_4$ molecules, might be instrumental in enhancing the earliest nucleation of silica oligomers, despite the occupancy of most interactions being low and the spontaneous condensation of silica being significantly slow.

### 3.3. MD Assessment of the SUMO-1 Systems

The small ubiquitin modifier-1 (SUMO-1) belongs to the family of ubiquitin-related proteins and is involved in the modulation of protein–protein interactions, as well as in post-translational modification processes (sumoylation). Moreover, it was observed that SUMO-1 plays a key role in the control of the nucleo-cytoplasmic transport, and suggested it might act in the regulation of the cell cycle and in cell apoptosis [19]. Despite sharing about 20% of the sequence with ubiquitin, SUMO-1 is associated with a ubiquitin-like core, involving a $\beta$-sheet and an $\alpha$-helix. In addition, SUMO-1 hosts three highly flexible domains, namely, a short carboxyl-terminus (GLN94 to VAL103), an extended amino-terminus (GLY1 to GLU22), and a small domain protruding from the ubiquitin core (GLN57 to ASN62 and PHE66 to GLU87, hereafter referred to as "middle" domain).

Firstly, we characterized the inter-molecular interactions between the residues of the ubiquitin-like core of SUMO-1, employing the PyInteraph toolkit (see the Materials and Methods section): Each entry of the heatmaps shown in Figure 4, reports the difference between the values of occupancy for each kind of intermolecular interaction between the control and the test trajectory (i.e., electrostatic and hydrophobic contacts, and hydrogen bonds). Overall, almost all entries display small differences, with very few exceptions, suggesting a negligible interference of $Si(OH)_4$ with the dynamics of the structured domain of SUMO-1—as similarly observed in the case of ubiquitin. Indeed, the observed discrepancies mostly involve residues either belonging to a motile moiety (e.g., a turn within a $\beta$-sheet) or lying close to a flexible domain.

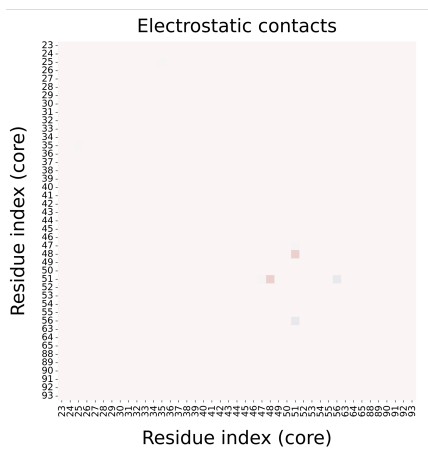 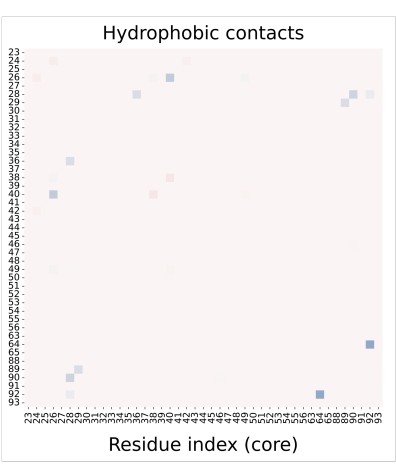 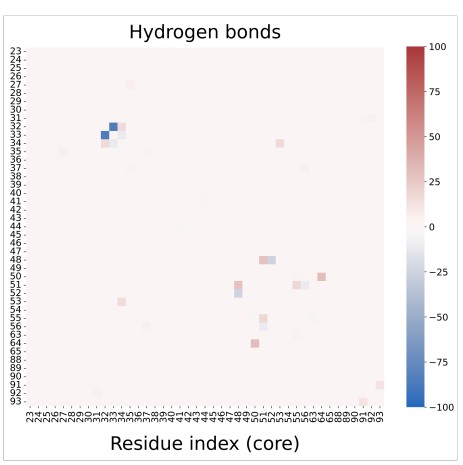

**Figure 4.** Heatmaps of the electrostatic and hydrophobic contacts, and hydrogen bonds, between the residues of the ubiquitin-like core of SUMO-1, obtained by subtracting the value of occupancy of each intermolecular interaction in the *control* and *test* scenarios.

We thus focused on the unstructured domains of SUMO-1, their characterization being less trivial on account of the carboxy- and amino-termini reversibly (and stochastically) adsorbing upon different portions of the protein surface in the two scenarios. This is in line with the expected behavior of both termini involving stretches of sticky, charged, and dipolar amino acids. In this concern, the assessment of the intrinsic flexibility of each residue of SUMO-1, based on the analysis of the $PAD_\omega$ factor (described in the Materials and Methods section), has been therefore limited to the first 325 nanoseconds of both trajectories, i.e., prior to the amino-terminus steadily adsorbing on the protein surface in the control scenario. The profile of the $PAD_\omega$ shown in Figure 5, shows multiple residues, mostly belonging to the unstructured domain of SUMO-1 (highlighted in red

in Figure 5), exhibiting a higher flexibility in the control rather than in the test scenario; this is corroborated by the RMSF profiles shown in Figure S4 of the Supplementary Data, suggesting a viscous buffering exerted by Si(OH)$_4$ on the mobile moieties.

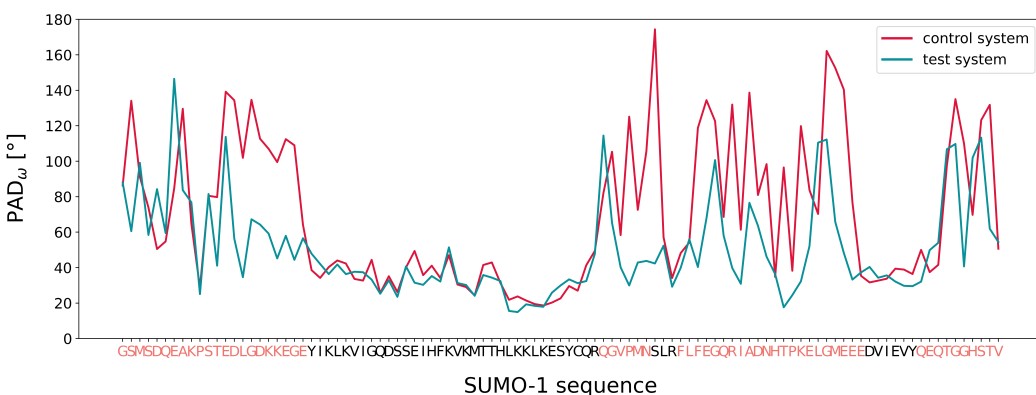

**Figure 5.** Profile of the PAD$_\omega$ on the first 325 nanoseconds of the *control* (red) and *test* (blue) trajectories of SUMO-1—residues belonging to unstructured domains are highlighted in red.

Indeed, as for ubiquitin, SUMO-1 is coated by a layer of Si(OH)$_4$ molecules (100 to 120 on average) establishing transient hydrogen bonds with one another and with the protein surface (see Figure 5 in the Supplementary Data). Yet, despite tracking numerous contacts between SUMO-1 and the orthosilicic acid, exceeding a 10% occupancy, no clear evidence of a putative binding pocket for Si(OH)$_4$ was detected, likely on account of the high flexibility of the unstructured moieties surrounding the ubiquitin-like core of SUMO-1.

### 3.4. MD Assessment of the IL22Rα1 System

Lastly, we focused on the behavior of the short (31 amino acids) transmembrane helix of the interleukin-22 receptor alpha-1 (IL22Rα1), here taken as a prototypical system of a hydrophobic protein region. As expected, the helix quickly unfolds (i.e., its overall helical content drops in both trajectories by 50% within 5 nanoseconds), collapsing into a diverse ensemble of closed conformations, yet never seemingly converging to an equilibrated structure, as shown by the RMSD profiles in Figure 6 of the Supplementary Data.

We thus assessed whether the unfolding of the helix shared similar conformational features in the two scenarios. On account of neither trajectory converging to an equilibrated basin, and on the helix rapidly transitioning between disordered conformations, we relied on a structural characterization based on the evolution of the secondary structure propensity of each residue, as described hereafter.

Briefly, each frame (that is, one frame every 0.5 ns from both trajectories) is assigned a string array, by associating each residue to a secondary structure element, i.e., 'H' for α helix, 'E' for β sheet, 'C' for coil, and so on. Thus, the Hamming distance between these strings, defined as the amount of different characters at each location of the array, is calculated, and employed as a similarity metric for a hierarchical clustering assessment of the frames upon an average-linkage criterion (see the Materials and Methods section). The output of this procedure is depicted as a dendrogram (shown in Figure S7 in the Supplementary Data), whereby we isolated five significant clusters from the merged trajectory.

Figure 6 reports the time distribution of the five clusters along the total trajectory, as well as the secondary structure occupancy of each residue within a cluster. Overall, a few patterns might be identified. Starting from an identical setup, the earliest frames in both the control and the test scenarios belong to a shared cluster (cluster **1**), as expected. From then onwards, the helix in the *control* scenario is allowed to reversibly inter-convert between states of moderate (cluster **3**), low (cluster **4**), and sparse (cluster **5**) helical content. Conversely, the trajectory of the helix in the *test* scenario is associated almost entirely with a unique cluster (cluster **2**), characterized by a small, fairly stable (between 40 and 60% occupancy) helical core, involving six amino acids in the middle portion of the

sequence—namely GLY17 to VAL22. As observed for the dynamics of the unstructured domains of SUMO-1, this might arise from a similar viscous buffering exerted by $Si(OH)_4$, seemingly dampening the flexibility of the mobile moieties, thereby constraining their conformational freedom.

We shall remark, however, that the observed discrepancies in the conformational clustering of the control and test scenarios (and the behavior thereof), might reflect the limited sampling afforded by 500-nanosecond MD simulations. Arguably, these differences might be further amplified by the potentially larger viscosity enforced by $Si(OH)_4$, such that longer or alternative MD replicas would result in diverse conformations of the transmembrane helix of IL22R$\alpha$1 being sampled.

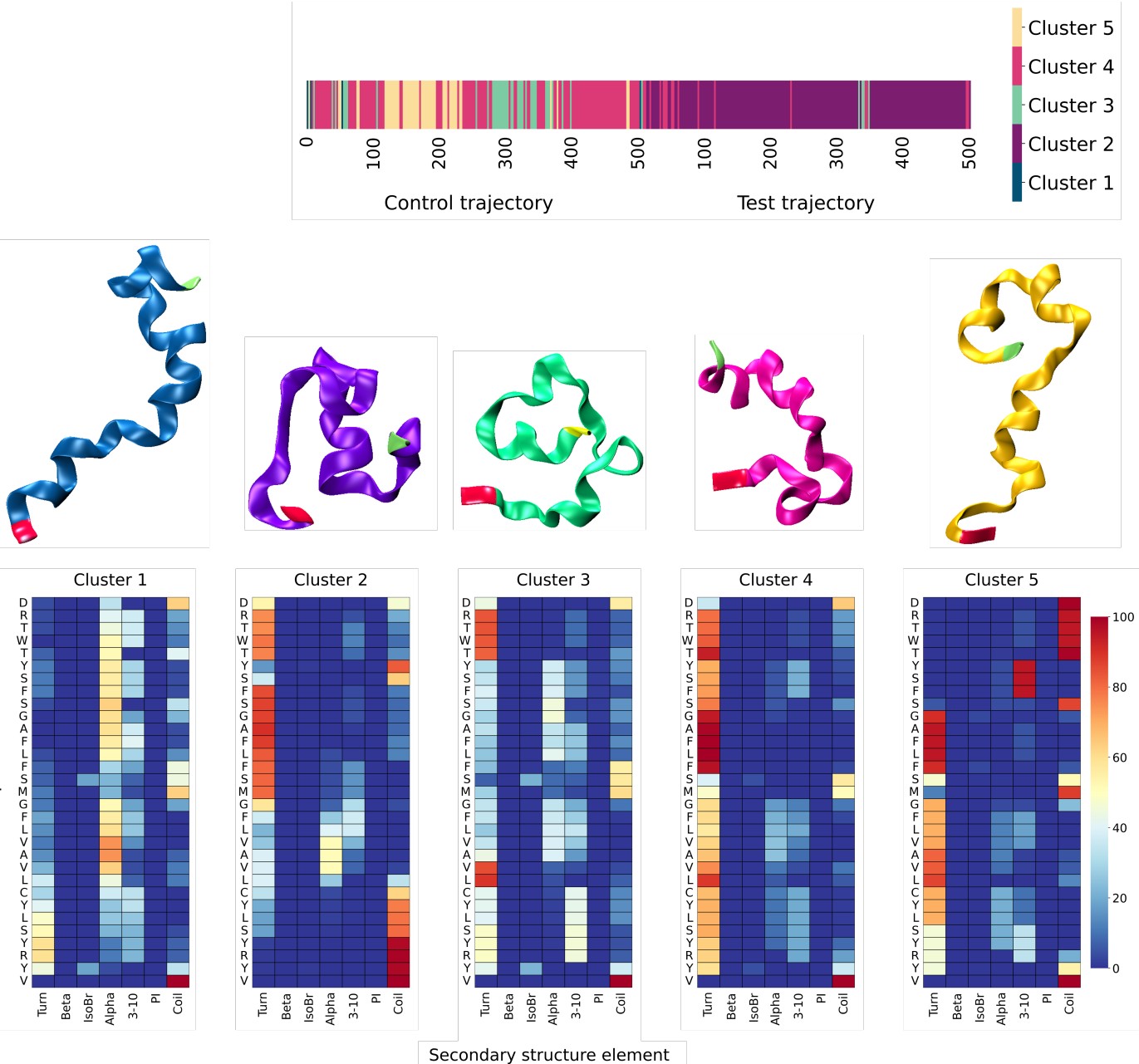

**Figure 6.** Top: Timeline distribution of the hierarchical clusters derived from the merged (*control*, *test*) trajectory of the transmembrane helix of IL22R$\alpha$1 (details in the main text). Bottom: Residue-wise occupancy of secondary structure elements and most representative structures associated with each cluster.

## 4. Discussion and Conclusions

The novel embedding protocol recently proposed by Sundell and co-workers [5], allows the application of atom probe tomography (APT) for the structural characterization of biomolecular systems. However, a few critical requirements need to be satisfied for the APT analysis to be effective: (i) the coating matrix must be chemically inert and exert no mechanical stresses on the specimen; (ii) the water molecules around the system must be effectively replaced by the matrix precursor—that is, a silica derivative, such as orthosilicic acid ($Si(OH)_4$)—to avoid losses in spatial and chemical resolution on account of local, thermal fluctuations; (iii) upon drying, the silica matrix must yield a malleable and glass-like structure, to be subsequently shaped in the form of a tip or a needle.

In this work, we established an all-atoms MD simulation protocol, aimed at depicting the earliest stages of the embedding by silica of diverse protein molecules. First, we characterized the equilibrium dynamics of a solution of water and $Si(OH)_4$, observing no significant aggregation phenomena between the molecules of orthosilicic acid within the sampled time scale (500 ns).

We thus assessed whether orthosilicic acid might affect the structural and dynamical behavior of small proteins in an effective manner. Indeed, $Si(OH)_4$ embeds all systems within a thin coating layer, with transient hydrogen bonds being established between the molecules of orthosilicic acid and with the surfaces of the proteins—albeit mostly in a non-specific way. Arguably, this emerging shell might enhance the condensation rates of the amorphous silica matrix, otherwise characterized by slow kinetics and typically catalyzed by means of prolonged curing in vitro.

Overall, we infer that $Si(OH)_4$ exerts a negligible (if any) interference on the behavior of structured, stably folded systems, such as ubiquitin and ubiquitin-like motifs, thereby acting as a properly inert molecule. Yet, the embedding by orthosilicic acid, seemingly buffers the dynamics of inherently motile and unstable moieties, constraining their conformational flexibility. While significant, these observations are inherently bound by a limited dataset, and therefore constitute a preliminary insight on the behavior of $Si(OH)_4$. In fact, it is likely that replicate simulations would achieve different conformations of the motile domains of SUMO-1 and, more significantly, more—or even different—conformational clusters of the transmembrane helix of IL22R$\alpha$1, here taken as representatives of an intrinsically disordered and of an unstable, hydrophobic moiety, respectively. Moreover, further scrutiny is required, e.g., by ranging the concentration of orthosilicic acid in the solution, which will be the object of a future assessment by the authors. Nevertheless, our characterization seemingly highlights effective (and highly desirable) features of orthosilicic acid as a silica precursor.

Another scenario worth exploring, is the step-wise synthesis of a silica matrix about a biological system upon drying, thus expanding on a recent work by Dupuis and co-workers, who proposed an in silico protocol combining the capabilities of an ad hoc reactive force field [37] with a grand-canonical Monte Carlo approach [38]. Further assessments based on numerical approaches, might aid in validating APT as an alternative to conventional crystallographic techniques, potentially overcoming known technical issues, by allowing the structural characterization of intrinsically disordered and flexible systems.

**Supplementary Materials:** The following supporting information can be downloaded at: https://www.mdpi.com/article/10.3390/biophysica3020018/s1.

**Author Contributions:** S.T. and G.L. oversaw and coordinated the whole research project. G.N.I. and F.C. performed the MD simulations. The data analysis was carried out by G.N.I., with the constant supervision of L.P. G.N.I. and L.P. mainly worked on the first draft, while the final manuscript was written with contributions from all authors. All authors have read and agreed to the published version of the manuscript.

**Funding:** We acknowledge helpful discussions with G. Marini, T. Morresi, and T. Tarenzi. This project was funded by the European Union, under Horizon Europe Programme (grant agreement 101046651—MIMOSA). The views and opinions expressed are, however, those of the authors only, and do not necessarily reflect those of the European Union or the European Innovation Council and

SMEs Executive Agency (EISMEA). Neither the European Union nor the granting authority can be held responsible for them.

**Institutional Review Board Statement:** Not applicable.

**Informed Consent Statement:** Not applicable.

**Data Availability Statement:** The raw data associated with this work are freely available on a Zenodo repository at the link https://zenodo.org/record/7733955#.ZBHFIezMI-Q.

**Conflicts of Interest:** The authors declare no conflict of interest.

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
