# Peer review of "Silica In Silico: A Molecular Dynamics Characterization of the Early Stages of Protein Embedding for Atom Probe Tomography"

_biophysica, doi:10.3390/biophysica3020018_

Round 1

Reviewer 1 Report

I find the paper well-written and providing solid results concerning formation of the internal structures of biomolecules in a water with orthosilicic acid solution. The method of molecular dynamics simulations used for the investigation is fully adequate to the problem being solved and reasonably applied. I believe that the paper may be published if the following small issues are addressed:

1. Caption to the figures must be self-consistent. In addition, all the information must be also contained in the paper text. Therefore, g(r) in the caption of Figure 1 must be explained. Moreover, the authors must clearly write that the coordination number of water molecules around a Si(OH)4 molecule is calculated based on the shaded area in Figure 1 not only in the Figure caption but also in the paper text.

2. Lines 226-227, page 8: I believe that the graphical abstract is to draw attention of a potential reader to the paper but is not a part of the paper content. Therefore, if the authors want to visually present in the paper text the information contained in the graphical abstract, respective figure(s) must be added.

Author Response

We thank the reviewer for his/her supportive assessment of the manuscript and for his/her comments.

  1. Both the caption to Figure 1 and the text in the main article (lines 141, 146, 149-150) have been modified accordingly. Likewise, the axis labels in Figure 1 were changed from "RDF" to "g(r)" for clarity.
  2. Indeed, it was a mistake on our side to believe that the graphical abstract would appear in the main article. All references thereof have been removed (lines 229-230).

Reviewer 2 Report

In this paper, Inverardi et al studied the effects of orthosilicic acid to the protein, including folded protein Ub and disordered protein SUMO-1. They propsed that the orthosilicic acid has few effects on folded protein but largely affect the disordered peptides. However, the work is unsoundness, which means their conclusions are made causually and arbitrarily.

1. The short simulation might not enough to characterize the effects of small molecules to the protein, especially for the protein with stable protein.

2. The authors compared the structural properties of proteins with or without Si(OH)4, they found that the structures are different for the disordered or flexible peptides. However, only one simulation runs was set as control or test system, respectively. If the authors will run multiple trajectories, they should find that the structural properties (like RMSF, contacts, secondary structures, etc) are also different in different runs of the same system. It is unable to conclude that differences are induced by Si(OH)4. They are intrinsical dynamics.

3. The figures should improved carefully, including the axies label and the presentation, like Figure 4.

Author Response

It was not easy to understand some of the comments prompted out by this Reviewer. However, we believe that some clarification is needed, since readers might be misled to the same conclusions.
1-2. We remark that this study is a preliminary characterization of the scenario that is commonly encountered in experiments with Atom Probe Tomography, namely the early-stage onset of an embedding layer about small biomolecules by orthosilicic acid. Hence the amount of data, as well as the analysis, is commensurate for that aim. Adding more trajectories on such a small system would not increase the quality of sampling, since the autocorrelation time (not reported in the manuscript) is in the order of 10 ns, at least for the stable regions of the studied proteins. The total simulation time of 500 ns is therefore sufficient to provide conclusions on the studied systems, except for the IL22RA1-helices, that clearly unfold in a hydrophilic environment.

3. The axis labels in Figure 4 have been modified accordingly. We thank the Reviewer for this comment.

Reviewer 3 Report

Please see my comments and suggestions in the attached document.

Author Response

We thank the reviewer for his/her supportive assessment of the manuscript and for his/her insightful suggestions.

  1. If by dimerization the reviewer refers to the supramolecular assembly of Si(OH)4, the peak in Figure 1A hardly depicts the close encounter between molecules of orthosilicic acid. In fact, Si(OH)4 is associated with a radius of approximately 2.5-3 Å, thereby establishing the minimum distance between the silicon atoms of Si(OH)4 molecules in close contact about 5-6 Å (i.e. lower than the observed peak by a factor of 2-3). Arguably, such peak might be accounted for by (mild) boundary artifacts in the estimate of the radial distribution function.
    On the other hand, (classic) MD simulations might not be appropriate to assess the dimerization/condensation process of Si(OH)4, unless associated with reactive force fields and/or ab initio techniques.
  2. We now show that the RMSF trends in both the control and the test ubiquitin scenario are in qualitative agreement with the experimental beta factors associated with the alpha carbons in the ubiquitin sequence (Figure 2A and lines 164-166).
  3. and 4. required a more elaborate response (see attachment). The legend of Figure 5 in the Supplementary Material has been corrected and its caption modified for clarity.

Round 2

Reviewer 2 Report

I guess the authors are new players to the molecular dynamics simulations and protein structure computation, that's why they can't understand what I meant. Actually, my questions are quit simple: the current results and data is insufficient to support their conclusions.

The major point in this paper is they observed no significant influences on stable protein induced by orthosillicic acid (OA), but largely affect the IDP.  What support their point is the RMSF/PAD value differences of the proteins with or without OA. But the intrinsical unstable nature of the unstructured domains (SUMO-1) makes the conclusions unreliable. If the authors could repeat the simulations of their control system and test system, they might find that the fluctuations of the SUMO-1 peptide are random, which means the fluctuation is large in some trajectories of OA-bound systems, but small in some trajectories of OA-free systems. That's why one need to run multiple trajectories to demonstrate the repeatability of their observations.

Besides, it should be molecular dynamics simulation, instead of molecular dynamics. They are totally different.

Even though the immature implementation and unsound conclusions of the current version manuscript, I'm glad to see the applications of MD simulations on this topic.  I hope  the authors could revise their manuscript carefully and throughly  in the next round.

Author Response

We thank the reviewer for clarifying his/her previous report and offering his views/criticisms. We have further analyzed our data and provide our reply here in a separate file.

We remark that the term "molecular dynamics" has not been used alone in our manuscript. We either write "molecular dynamics characterization" or, as pointed out by the reviewer "molecular dynamics simulation" or we specify that it is a numerical approach. 
